# Stimulus-Organism-Response Framework: Is the Perceived Outstanding Universal Value Attractiveness of Tourists Beneficial to World Heritage Site Conservation?

**DOI:** 10.3390/ijerph20021189

**Published:** 2023-01-09

**Authors:** Sifeng Nian, Donghe Li, Jie Zhang, Song Lu, Xiaowan Zhang

**Affiliations:** 1Center for Hui Studies, Anhui University, Hefei 230039, China; 2School of Business, Anhui University, Hefei 230601, China; 3School of Geographic and Oceanographic Sciences, Nanjing University, Nanjing 210023, China; 4School of Environmental and Geographic Sciences, Shanghai Normal University, Shanghai 200233, China

**Keywords:** sustainable development, world heritage site (WHS) conservation, Outstanding Universal Value (OUV), destination attachment, tourist value, Stimulus-Organism-Response (S-O-R)

## Abstract

Tourists have been attracted to world heritage sites (WHSs) by their Outstanding Universal Value (OUV). In view of the Stimulus-Organism-Response (S-O-R) framework and the theory of attitude and behavior, by employing 563 tourist samples from Mount Sanqingshan National Park, and using structural equation modeling, we examine tourist behavioral intention for heritage conservation, and the following conclusions were drawn: (1) the S-O-R theory revealed the behavioral intentions of tourists to protect WHSs; (2) as a stimulus, tourists’ value perception and destination attachment were positively affected by the OUV attractiveness, and their perceived value had a positive influence on heritage conservation, although the hypothesis of destination attachment to heritage conservation was not supported; (3) heritage-conservation education and knowledge positively influenced tourists’ behavioral intentions towards heritage protection, and tourists’ heritage protection attitude had a positive influence on their behavioral intention; and (4) a framework of the influence mechanism for tourists’ heritage conservation based on the S-O-R theory was proposed, while tourists’ cognitive and affective attitudes impacted on heritage protection intention which, in turn, further enhanced the tourists’ perception of the OUV. Conclusively, the measures and implications were proposed for improving conservation and management of WHSs, in particular to achieve the sustainable development of the tourist industry and world heritage sites.

## 1. Introduction

UNESCO and the World Heritage Committee recognize WHSs as precious and irreplaceable treasure, as part of a cultural heritage and natural landscape acknowledged by all of humanity with outstanding significance and universal value, along with their display, scientific, inheritance, and existence values [1]. This is also the basic requirement of the convention concerning the conservation of the World Cultural and Natural Heritage, and the OUV is the core criterion and the most valuable basis for nominations to become WHSs [1]. The status of a WHS is not only as a place needing most valuable protection, but it is also a hot spot with iconic value for tourism and for regional or economic development [2,3]. In particular, the tourism value of WHSs is used to develop tours and sightseeing and popularize science education. The WHS is a world-class ‘famous brand’; once a place has been successfully nominated, the ‘golden signboard’ will become a famous tourism destination both nationally and globally, and in particular, the WHS nomination will usher in the rapid development of tourism. By July 2021, the overall amount of WHSs had reached 1122, dispersed throughout 167 nations. China is a developing country with numerous WHSs (with 56 WHSs, China, along with Italy, has the most world heritage projects.). In accordance with the World Heritage Convention, projects with display, scientific, cultural, and ecological value have been performed in WHSs. Conversely, the large benefits brought by the ‘money tree’ of WHSs has been accompanied by huge pressure and even threats triggered by the rapidly expanding tourist industry, which is an especially serious problems faced by the tourist development of world natural heritage sites (WNHSs) [4]. Because a nominated WHS is a nationwide ‘tourist highlight’ or ‘must-visit’ attraction, it is deemed a magnet for tourists [5]. Just as Caust and Vecco (2017) wonder, is it a blessing or a burden to be listed as a WHS in developing Asian countries [6]? In particular, China’s huge population (1.41 billion people in mainland China in 2021) and its rapid economic development since its reform and opening up (from 1978 to 2018, the average annual growth rate of GDP was 9.5%) have generated considerable demand for leisure tourism (in 2019, the number of domestic tourists reached 6.0 billion), producing the rapid growth in the volume of international tourists in recent years. As an outcome, the number of tourists visiting WHSs has skyrocketed. For example, Mount Sanqingshan National Park (MSNP) has seen its tourist numbers soaring from 1.48 million in 2008 to 23.95 million in 2019, following its successful WHS application in 2008. This undoubtedly put enormous pressure on the protection of heritage sites. Generally, WNHSs are ecologically fragile and vulnerable to ecological pressure and threats. Chinese world heritage sites often face issues such as runaway tourist capacity, overdevelopment of tourist facilities, serious over-commercialization and adverse impacts, which pose major challenges to conversation and sustainable development of WHSs [7,8,9,10,11].

WHSs have the characteristics of being scarce, non-renewable, non-replaceable, and needing long-term protection, so protection is a core issue of tourist industry concerning WHSs. Academics have primarily researched the association with WHS conservation and tourism development from the perspective of government behavior, organization strategy, community involvement, and the performance of other stakeholders [12,13,14,15]. Buckley (2018) believed they have a multifaceted connection between tourist development and the preservation of a world heritage area, and suggested that tourism businesses could become advocates for the preservation of heritage sites, but there is no proof that most tourism customers are such advocates [16]. However, because the large number of tourists form the main part of tourism activities at WHSs, their attitudes and behaviours will have a certain influence on the protection and sustainable development of WHSs. Tourists, as important stakeholders in WHS protection, have a direct feeling and influence on the ecological, psychological, facility, and social capacity of WHSs. Tourists are the agents of ecological, environmental, low-carbon, and sustainable tourism, as well as being the negative influences of various tourism forms. Heritage tourism is seen as a special tourism experience, with a feeling of awe that may be able to motivate tourists to actively protect WHSs. Tourists are not only the beneficiaries and experiencers of heritage protection, but also participants, executors, and contributors to preservation as well as supervisors and contemporary witnesses. How do tourists feel about an OUV attraction when they experience the value of heritage? How do education and knowledge relate to heritage protection and affect their OUV perception and heritage conservation? What are the attitudes and behaviors of tourists when acting as stakeholders and custodians of heritage conservation? What factors motivate their internal state and emotional changes, as they experience heritage conservation and heritage tourism development? How do these factors translate into action? How do they respond to their intention and behavior, and how much do these factors influence and change their attitudes and behaviors towards heritage protection? These are the starting points and emphasis of this study.

Much attention has been paid to tourists’ attitudes and behaviors towards heritage protection and related influencing factors. From the perspective of the attraction of a WHS’s OUV, its value was emphasized [17,18,19]. From an active brand marketing perspective, the world heritage title could entice tourists and generate shared awareness, and the OUV attraction plays a key position in tourist appreciation and loyalty to tourism destination [18,20]. The all-encompassing appeal of the WHS has a significantly positive influence on tourist place and destination attachment, which helps to promote heritage protection [21,22,23,24]. Researchers have examined the factors influencing and the connection among tourist place, heritage, and destination attachment for the protection and preservation of WHSs [25,26,27,28,29]. Furthermore, studies have been conducted from the perspectives of tourists’ perceived value, experience and service quality, satisfaction, environmentally responsible behavior, and world heritage protection awareness [28,30,31,32,33,34,35]. Tourism is an experiential activity and an emotional process, tourists’ satisfaction and loyalty have a positive emotional influence on heritage protection [36,37]. Some scholars also studied tourists’ attitudes towards protecting cultural heritage from the perspective of tourists’ environmentally responsible behavior, and ecological and sustainable tourism [25,26,38,39]. Through the interpretation and popularization of the OUV, heritage protection education can inspire most stakeholders, especially the public, to value heritage so as to better protect the natural world and their cultural heritage. Tourists’ heritage protection education and knowledge have a certain impact on their attitude and behavior towards heritage protection [40,41]. In particular, heritage protection education aroused persons’ awareness of world heritage conservation, generated environmentally responsible behavior [42], as well as changing tourist emotions, satisfaction, and behavioral intentions, promoting heritage conservation [37].

Research on the attitude and intention of heritage protection mainly involves the following theories: theory of planned behavior (TPB) [22,43], Value-Belief-Norm [44], place attachment [23,45], perceived tourism value [30,46], and visitor experience and resource protection [47,48]. To the best of our knowledge, the Stimulus-Organism-Response (S-O-R) framework has not been adopted to investigate tourists’ attitudes and behaviors towards WHS conservation. The S-O-R theory aims to reveal how people change their internal state when stimulated by external environmental factors, so as to make a behavioral intention response [49]. Taking a relatively overall multidimensional perspective, we adapted the S-O-R framework, using tourists’ OUV attraction and heritage conservation education as the Stimulus (S), perceived tourism value and destination attachment as the Organism (O), and heritage conservation attitude and behavioral intention for the Response (R). We constructed the conceptual framework based on the tourist perception of heritage preservation, utilizing structural equation modeling (SEM) as the methodology and the MSNP as the study area, to explore the correlation among tourists’ attitudes and behaviors towards heritage protection, and to further explain the affecting determinants and mechanism of visitors’ WHS conservation, so as to benefit heritage protection and sustainable tourism development.

In the following sections, we provide an overview of the S-O-R framework, review the concepts of relevant core dimensions, and propose relevant research hypothesis. The next section introduces the research method and data analysis. Finally, we discuss the results and put forward implications, and provide study limitations and future research prospects.

## 2. Conceptual Framework and Hypothesis

### 2.1. Stimulus-Organism-Response Framework

The S-O-R framework was created by Mehrabian to reveal how people respond to an external behavioral stimulus (S), forming a change in his/her personal internal state (O), and triggering his/her personal response (R) [49]. The internal state is a medium and mediation of vitro stimulation and eventual response. Stimulus can trigger a person’s cognitive and affective state, in order for them to decide whether to adopt approaching or avoiding behavior [50]. Stimulating factors can include subject and psychosocial stimulation [51]. Subsequent researchers further believed that objective and psychosocial stimuli triggered individual cognitive and emotional states, leading to individual behavioral tendencies and psychological outcomes [52]. The S-O-R theory has been widely applied effectively within various fields such as service, consumption, tourism, and environmental protection; therefore, it is an important analytical framework to explain people’s behavioral processes [50,53,54,55,56,57,58,59]. Within the context of the tourism experience at WNHSs, the stimulus included the OUV attraction, tourist heritage conservation education, and other factors related to service quality. This kind of experience and stimulation triggered tourists’ inner feelings, such as value perception, destination attachment, etc., prompting tourists to produce corresponding behavioral responses and psychological results. The S-O-R theory was adopted in this research, using the attraction’s perceived OUV and tourists’ education and knowledge of heritage protection as incentives; perceived tourism value and destination attachment were the intrinsic organism, and were explored to promote or weaken the tourists’ behavioral intention response towards heritage protection and to explain the influencing factors and mechanism of WHS conservation.

### 2.2. Stimulus: OUV Attraction and Conservation Education

#### 2.2.1. Perceived OUV Attraction

The OUV is related to cultural and/or natural attractions of importance so extraordinary that it exceeds national limits, and is of such significance that it represents all upcoming generations of humankind [1]. Being nominated as a WHS aims to protect the common property of humankind, which is of great significance to the entire international community. The WHSs’ OUV is the chief foundation for its nomination, including 10 assessment standards, with integrity and/or authenticity, sufficient preservation and management mechanisms that ensure WHS is sheltered and could be deemed an OUV [1]. The WNHS refers to a natural resort or an obviously described natural zone with OUV from the view of science, conservation, and natural beauty. The OUV is the maximum direct manifestation of heritage charm and essential attractiveness for tourist growth, which is also the key effort of WHS preservation [3,60]. Travel resort charm is a main feature of tourist academic research, whether it is from perceived emotional experience or functional service perspective, the OUV is certainly the center rank of the site [19]. Destination attractiveness refers to how a destination meets a person’s needs or personal perceived benefits and consists of core and extended attributes [61]. In a WHS, core qualities relate to distinctive natural and humanistic capitals, just as OUV, while extended ascribes refer to functional features, such as well-behaved tourist amenities, care services, and highly effective organizational management. In general, the two complement each other and jointly enhance a heritage site’s attraction to tourists [62].

Employing USA WHSs as case research, Hazen (2009) examined the perception measurement of tourists’ OUV, showing that the aesthetic, cultural, educational, environmental, reactional, and spiritual value aspects of WHSs could improve its heritage conservation [18]. Baral et al. (2017) adapted Mt. Everest National Park to measure the OUV in WHSs and the dimension of perceptions of tourists to its OUV, containing importance, uniqueness, influence, legacy, values, and appeal [63]. Investigation had shown that an OUV attractiveness had a key impact on tourist loyalty to the place, experience of value perception, environmentally conscious behavior, and attitudes and behavior towards heritage protection [23,24,29,64,65]. Even the attraction of ordinary tourist destinations had a significant positive impact on tourists’ destination attachment [21]. A destination’s main attraction and its secondary attractiveness, just as in the case of tourism services and native communities, had positive influences on tourist resort attachment, experience of perceived tourist values, and tourists’ attitudes and behaviors towards defending tourism destination [66]. Research by Reitsamer et al. (2016) demonstrated that embodied perceptions of tourist resort attractiveness could improve destination attachment and sensed tourism values, change visitor’s attitudes, and thus improve visitors’ satisfaction [67]. The attractiveness of a WHS has a positive impact on visitor motivation, influencing the tourist’s feelings and providing pertinent wisdom, which could stimulate and transform the inner state of visitors. Destination attractiveness could intensify tourist place attachment and endorse environmentally responsible behavior as well as increase tourist effectiveness and visitor satisfaction [68]. As a core charm of WHSs, the OUV has a valuable impact on how visitors experience value, tourism motivation, tourist satisfaction, and connectedness with the destination. Consequently, the following hypothesis were proposed:

**Hypothesis** **1** **(H1).**The OUV attraction has a positive impact on the perceived touristic value.

**Hypothesis** **2** **(H2).**The OUV attraction has a positive impact on the destination attachment.

#### 2.2.2. Heritage Conservation Education

World heritage education is an important way to understand and protect WHSs. After UNESCO staged the convention concerning the protection of the world cultural and natural heritage in 1972, heritage education has received global attention and countries have developed relevant policies and implementation plans for heritage education. Heritage education and preservation, and sustainable development of WHSs, are supported by a wide range of stakeholders, including governments, local communities, tourists, and NGOs [69]. Environmental education was defined by the World Conservation Union (IUCN) in 1970, in which knowledge transforms behavior at personal, societal, and global levels so that people may acquire the skills needed to recognize and solve environmental issues. Through the interpretation and publicity of the value of WHSs, the popularization of science and education in relation to heritage stimulates the attraction of many stakeholders, especially the public, making them wish to better safeguard WHSs [70]. WHS protection education raises a human being’s awareness of heritage conservation and generates environmentally conscious behavior that has a positive influence on heritage conservation [71]. Cottrell (2003) pointed out that education can change a person’s cognitive (professed knowledge of environmental problems), affective (environmental concern), and conative (verbal commitment) behaviors, resulting in pro-environmental behavior intentions [72]. Knowledge of UNESCO’s WHSs can affect tourists’ emotions, satisfaction, and behavioral intentions [37]. Other scholars have studied the influence and role of environmental knowledge of and sensitivity towards heritage protection [41,73,74], as well as association among environmental conservation and sustainable development [40], and developed the environmental conservation knowledge questionnaire scale [75]. According to this review, heritage protection education had a positive impact on tourists’ conservation intentions and environmental conservation behavior intentions. Thus, the following hypothesis were proposed:

**Hypothesis** **3** **(H3).**Heritage conservation education has a positive influence on heritage conservation attitude.

**Hypothesis** **4** **(H4).**Heritage conservation education has a positive influence on heritage conservation intention.

### 2.3. Organisms: Perceived Tourism Value and Destination Attachment

#### 2.3.1. Perceived Tourism Value

In the tourist research community, perceived value was frequently conceptualized as the individual assessment of the traits of tourist products, including service quality, price, emotions, and social factors, which determined whether the value was worth consuming and whether it could influence satisfaction after travelling [76]. Perceived tourism value was the sum of tourism motivation and tourism experience value perception. Lu (2016) showed that tourism resource quality perception was the primary factor affecting environmental responsible behavior, followed by tourism service quality and tourism activity experience, and noted that satisfaction especially had a positive influence on the promotion of tourism value and environmental protection behavior. Perceived value was an important predictor of tourist behavior [36]. Chiu et al. (2014) observed that while tourists feel value for money is important, they have optimistic feelings about the destination and their behavior benefits the destination [77]. Williams and Soutar (2009) divided the perceived value into four dimensions: functional, emotional, social, and novelty, and signified that it had a positive impact on tourist satisfaction and behavioral intentions [33]. Prebensen et al. (2013) applied tourist motivation and involvement as antecedents to the perception of destination-experience value, which was divided into a functional, social, and epistemic value [31]. Chen et al. (2016) showed that symbolic, experiential, and functional consumption have positive influence on destination attachment and tourist satisfaction and can promote heritage-protection behavior [30]. Chen and Chen (2010) examined the correlation among experience quality, perceived value, satisfaction, and behavioral intentions of visitors at heritage sites, and suggested that perceived value can increase satisfaction and change tourist behavior [78]. Su and Qian (2012) took the Chinese classical gardens from among the world cultural heritage sites as an example, indicating that tourist-perceived tourism functions have a positive impact on WHS protection [23]. Tang et al. (2007) took Jiuzhaigou WNHS as an example, revealing that tourist experience gives the destination special meaning and value in terms of emotions, knowledge, recreation, and spirit, and that in turn helps tourists create a sense of place that is far more conducive to heritage conservation and displaying heritage values. In summary, the tourist-perceived tourism value had a positive influence on heritage protection and environmental behavioral intentions [29]. Thus, the study proposed the ensuing hypothesis:

**Hypothesis** **5** **(H5).**Perceived tourism value has a positive influence on heritage conservation attitude.

**Hypothesis** **6** **(H6).**Perceived tourism value has a positive influence on heritage conservation intention.

#### 2.3.2. Destination Attachment

Tuan (1977) points out that sense of place is the characteristic of place itself and human’s affection to location, noting that the subsequent development of experience, memory, and intention leads to the formation of a deep attachment to place, which is referred to as place attachment [79]. Williams (1992) proposed that place attachment contains both place identity and place dependence, and the former is a functional attachment between humans and places, and the latter is an emotional attachment [80]. The notion of place was together physical and psychological, and was interpreted, perceived, understood, and imagined in this narrative [81]. Tourism and travel are key ways for people to sense and comprehend the setting as an intersection of themselves and the place, with significant representative meaning for visitors. Ramkissoon (2013) classified place attachment into place dependency, place affect, place identification, and place attachment, indicating that the four dimensions of place attachment significantly and positively influenced place satisfaction and environmental performance [82]. The sake of the sense of place study was to examine the importance and value of the tourism destination’s amenities. Tang et al. (2007) illustrated that being tied to a place has a positive influence on resource protection and heritage preservation [29]. Using the example of the Chinese classical garden of WHS, Su and Qian (2012) demonstrated that place attachment has a significantly positive influence on the attitudes and behaviors of tourists towards to heritage protection [23]. Williams and Patterson (1999) considered that people’s psychological affection towards an environment will inspire them to engage in additional responsible environmental behaviors, for example taking the initiative to pick up garbage and respecting animals [80]. Relevant research has shown that attachment to tourist destinations has been an important precursor variable of tourism environmental protection and heritage-protection intentions, which can change tourist behavior intentions and have positive impacts on heritage protection [28,56,83]. Destination attachment had a positive influence on tourist satisfaction, and cognitive, affective, and conative loyalty in Yuksel et al. [84]. The local connection was an important basis for tourism’s environmental protection intentions and had a significantly positive influence on tourists’ environmental behavior and their appreciation of travel destinations [26,56,68,73,85,86,87]. Williams and Vaske (2003) used the generalizability of a psychometric method to gage place attachment efficiency and showed that tourists have a better comprehension of the destination, and that heritage can produce positive psychological outcomes [88]. Thus, we proposed the following hypothesis:

**Hypothesis** **7** **(H7).**Destination attachment has a positive influence on heritage conservation attitudes.

**Hypothesis** **8** **(H8).**Destination attachment has a positive influence on heritage conservation intentions.

### 2.4. Response: Heritage Conservation Attitude and Intentional Behaviour

As one of the key stakeholders, tourism environmental protection behavior and heritage protection attitude played a crucial position in heritage preservation and tourism sustainable development. Tourists’ environmentally responsible behavior was the beginning point for WHS protection. It referred to the behavior in which tourists had the least unenthusiastic influence on the ecological setting and promoted the sustainable application of resources in the tourist resort, also recognized as pro-environmental behavior, eco-friendly behavior, and ecological behavior, which was also a crucial aspect of heritage safety [25,89,90]. In view of the particularity of tourist activities in heritage sites, Chiu et al. (2014) considered visitor’s attractive, cognitive and emotional imagery, perceived tourism value, leisure participation, perseveration obligation, and environmental direction as key components affecting green tourist behavior [77]. By interpreting and popularizing the OUV and other elements of a heritage site, heritage education arouses the interest of a bulk of stakeholders, and notably the public, to do more to defend the world’s natural and cultural heritage [91]. Heritage conservation education has impacted humankind’s awareness of heritage preservation and has produced environmentally responsible behaviors [71].

Stern (2000) proposed the VBN theory that expanded investigation elements of environmental behavior and has been verified and applied in numerous studies [44]. Researchers primarily examined the correlation and contributing determinant of environmental behavior and heritage preservation attitude intention from the view of the attitude behavior theory, under which the TPB had been commonly used. According to the TPB, the actual behavioral intention of human beings was influenced by attitudes, social norms, and perceived behavioral control [43]. Scholars such as Hu et al. (2019) have used TPB to study tourist environmental behaviors and heritage protection intentions in WHSs, suggesting that attitude had a significant impact on behavioral intention [22]. In view of the highly contextual characteristics of tourist environmental responsible behaviors, based on the rational behavior theory and the TPB, the Stimulus-Organism-Response (S-O-R) framework proposed by Mehrabian A (1974) [49] may be an effective tool for exploring environmental behavior and world heritage conservation mechanisms. This theoretical framework has been applied widely in the researching of human behavior. It has good predictive power and has been named as the environmental stimuli-emotional states-behavioral responses theory by many scholars. Su and Swanson (2017) used the S-O-R framework to study the impact of destination social responsibility on the environmental responsibility behavior of tourists, indicating that this model was suitable [57]. Tourist activities were especially part of the process of emotional experience, and emotion had an important influence on tourist intention [46]. Therefore, we adapt the S-O-R outline to analyze visitor heritage protection attitudes and behavioral intentions and proposed the following hypothesis:

**Hypothesis** **9** **(H9).**Heritage conservation attitude has a positive influence on behavioral intention.

### 2.5. Proposed Conceptual Model

Based on the S-O-R theory, we constructed a conceptual model based on tourists’ OUV attraction perception and heritage protection education as the stimulus (S), perceived tourism value and destination attachment as the internal state (O), and heritage conservation attitude and behavioral intention as the response (R). The suggested conceptual model is portrayed in Figure 1.

## 3. Methodology

### 3.1. Study Area

Mount Sanqingshan National Park (MSNP) is situated in Jiangxi Province, PRC, which is named as the three peaks of Yujing, Yuxu, and Yuhua, similar to the three supreme Taoist gods of Yuqing, Shangqing, and Taiqing, who are seated on the mountain (Figure 2). The earliest development of MSNP tourism began in 1980, and major new tourist hotels and other tourist reception facilities began around 2000. In 2005, investment projects and development intensity increased. MSNP was awarded the WNHS title by UNESCO in 2008, was classified the state AAAAA tourist area as the national top-level tourist destination in 2011, and was granted world geopark status in 2012. UNESCO’s World Heritage Committee remarks that MSNP features a distinctive set of forested, fantastically shaped granite pillars and peaks concentrated in a fairly small region. Its OUV is exceptional, and also a well-known Taoist cultural site with more than 1600 years of history. As one of the greatest remarkable natural sceneries, MSNP has attracted many domestic and international tourists from countries such as Korea and Japan every year. The variety of tourists in MSNP in 2002 was 580,000 and the tourism revenue was 213 million RMB, but they increased to 23.95 million and 22.22 billion RMB, respectively, by 2019 (http://www.zgsr.gov.cn/sqs/bindex.shtml, accessed on 9 September 2019). However, the fast enlargement of visitors has started to put stress on MSNP, containing adverse ecological and environmental impacts, tourist accumulation, and other heritage preservation problems. Therefore, the selection of MSNP as the subject of study on tourist OUV attraction, destination attachment, heritage conservation, and sustainable development had a certain typicality and enabled MSNP to be an ideal investigation area.

### 3.2. Measurement Instruments

The questionnaire comprised two parts. The first part addressed the demographic traits. The second part consisted of four dimensions and twenty-six items constituting the proposed conceptual model (Table 1) and mainly from the literature research and the preliminary investigation of the case by the project team in May 2015. The OUV attractiveness (five points) was obtained mainly from related literature [17,18,66,68], and the criterion (vii) of the WHS, official website, and other related materials for application in WHSs, especially regarding the current situation of MSNP, such as the high-altitude walking trails (more than 20 kilometers), although it was not part of the OUV; however, the OUV charm that can be better displayed was an important medium for tourists to perceive OUV, and we use it as one of the items measured using OUV attraction. We paid attention to an OUV assessment that aggregated the global context with the local background. Source of measurement items for the conservation education (four items) was chiefly taken from the relevant literature and documents of the heritage preservation regulations [1,37,40,72,75,92]. Perceived tourism value (five elements) was mainly obtained from [31,33,78,93]. Destination attachment (four elements) was taken from relevant literature [23,82,84,88,94,95]. Heritage conservation attitude (three elements) was mainly obtained from the related literature [1,23,43,90,96,97]. Heritage conservation behavior (five elements) was mainly drawn from related works [1,23,43,45,90,96,97]. Scoring for each declaration was based on a Likert scale ranging from (1) totally disagree to (5) strongly agree.

### 3.3. Sampling and Data Collection

The questionnaire is distributed by the team members participating in survey design and release exercising. The field investigation was performed in August 2015 and the surveys were delivered at the key arrivals and departures Jinsha Ropeway and Waishuang Ropeway in MSNP. In order to promote the excellence of the survey and recovery rate, we adopted the procedure of setting up the two main export outlets in this scenic resort and giving souvenirs as gifts, choosing tourists after the trip to MSNP. According to convenient sampling methodology, the investigation group dispensed a total of 610 questionnaires and recovered 588. Excluding those that were non-severe, incomplete, or illogical, a total of 563 valid surveys were observed, yielding an effectual response of 90.8%. The study team surveyed stakeholders such as visitors, scenic destination managers, tourist practitioners, and native communities on questions about heritage preservation and tourist development to get a profounder insight into WNHS conservation.

### 3.4. Data Analysis

This research applied confirmatory factor analysis (CFA) using structural equation modeling (SEM). SEM boosts create, estimate, and analyze the causality model and is extensively utilized in social-science study. The investigation performed an examination of the measurement model, evaluated its validity and reliability, using SEM to identify correlations between the latent constructs. SPSS 22.0 statistical software was applied to correlate the sample databases and between the CFA and SEM methods were performed using AMOS 19.0 (SPSS Inc., Chicago, IL, USA). SEM was employed to check whether the model parameters had an inverse estimation hypothesis and to review the theoretical model to visualize whether there was any common method bias (CMB).

## 4. Results

### 4.1. Sample Profile

The example summaries are given in Table 2. The percentage of men and females was almost equal. The proportion of young people under 30 years old has reached nearly 60%. Students account for nearly 40%, which is related to the fact that summer vacation is the peak time for students to travel. The levels of education were high school and college graduate (43.4%), and university graduates and above (37.7%). The average monthly income was less than 1500 yuan, accounting for 37.8%, which is related to the students’ non-income. Types of tourism included participation in tour groups (33.7%), trips with family or friends (41.2%). Tourist incentive was chiefly natural sightseeing (72.3%), which was consistent with mountain-type natural attractions.

### 4.2. Reliability and Validity Analysis

Reliability testing is designed to check the reliability, stability, and consistency of scale data. Higher reliability indicates a slighter standard error of measurement. Using SPSS 22.0 to test, it was demonstrated that the general robustness of the Cronbach’s alpha scale was 0.95, and that the confidence coefficient of the Cronbach’s alpha of every dimension was above the threshold of 0.7. The comprehensive reliability of every dimension is greater than 0.8, signifying that the model is reliable [98]. The sake of the validity testing is to assess the validity of measures scale. The Kaiser–Meyer–Olkin (KMO) value of the total sample was over 0.9, which indicated the reliable construct validity of the questionnaire. The KMO of the content validity was greater than 0.7, suggesting that the developed measurement clauses could designate the substance to be measured. The average variance extracted (AVE) greater than 0.5, signifying that the observed variables could gage the latent variables, and that the convergent validity of every structure is suitable [99] (Table 3).

In addition, the value of the variance inflation factor (VIF < 10) demonstrated the absence of multicollinearity in this research [100]. The results in Table 4 indicated that the association coefficients were smaller than the square root of AVE, demonstrating that the discriminant validity was reasonable [99]. In conclusion, all indicators are reasonable, and the data gathered was satisfactory.

### 4.3. Descriptive Statistical Analysis

Tosun (2002) established the average score on the five-point Likert scale: between 1.0 and 2.4 is objective, 2.5 to 3.4 is neutral, and 3.5 to 5.0 is agreement [101]. Therefore, the OUV attractiveness score (4.39) is high, suggesting that OUV is the central driver and main attraction of heritage tourism. The average score (3.86) of heritage conservation education showed that the visitors’ perception was not high, and this may be connected to the insufficiency of Chinese WHS education. The higher mean score of perceived tourism value (4.22) indicates that visitors are satisfied with the value of heritage tourism. The mean score of destination attachment (4.01) is high, indicating that visitors have some perceived and emotional attachment to heritage tourism destinations. The relatively high mean scores for WHS conservation attitudes (4.36) and behavioral intentions (4.21) indicate that visitors are more willing to engage in the ecological safeguarding and preservation of WHSs.

### 4.4. Confirmatory Factor Analysis

#### 4.4.1. Testing of Measurement Models

The structural model was tested and analyzed to investigate the correlation between the variables in the measurement model. At first, the multivariate normal distribution of the sample was examined. The absolute value of the noticed variable of skewness (0.596–1.847) was smaller than the doorsill of 2.58; the absolute value of the kurtosis (0.272–4.000) was smaller than the doorsill of 10, so the sample data could be considered as having multivariate normality distribution. Additionally, CMB was examined. Exploratory factor analysis was presented, adopting Harman’s single-factor test [102]. The first factor clarified 19.04% of the overall variance, signifying that CMB was not a serious problem and that it can be ignored. Thirdly, while checking the total model fit indicators, Hair et al. (2002) advocated checking for whether there is a violation of the model parameter estimation, which can start in two ways: either there is negative variance of the error or the standardized parameter coefficient is weightier than or equal to 1 [100]. The error variance ranged from 0.033 to 0.046, with no negative error variance in the model. The standardized parameter coefficient ranged from 0.691 to 0.876, and was smaller than one, signifying that no violation of the estimation was able to impact the goodness-of-fit of the model. Ultimately, we adopted the maximum likelihood technique to evaluate the parameters of the theoretical model. Nunkoo et al. (2013) discovered that the pertinent fitting parameters are not perfect, so we intend to formulate further adjustments to the conceptual model [103]. Based on the measurement model of SEM, it is possible to correlate potential variables and establish relationships between potential variables. The correlation coefficient of OUV attraction and conservation education was 0.61, reaching a significance level of 0.001. The modified structural-model-fit indices were relatively appropriate: chi-square degrees of freedom (CMIN/DF) equal to 3.2, GFI equal to 0.879, RMSEA equal to 0.063, IFI equal to 0.937, TLI equal to 0.927, CFI equal to 0.937 in Table 5 [104]. Furthermore, for the revised model, the CMIN/DF were above the doorsill of three, and the remainder of the indicators attained a suitable value. According to Mulaik (2007), the ratio of the CMIN/DF is smaller than 5 when the sample size is greater than 500, instead of the usual 3, so the CMIN/DF ratio of 3.2 is satisfactory [105].

#### 4.4.2. Testing Structural Models

The hypothetical causal correlations were checked, and the outcomes of the assessment were presented in Figure 3.

## 5. Discussion

### 5.1. Factors Influencing Tourists’ Perception of Heritage Conservation

We found that the hypothesis of H1, H2, H3, H4, H5, H6, and H9 were supported. However, H7 and H8 were rejected.

#### 5.1.1. Effect of OUV Attraction and Conservation Education

As an important stimulus for tourists’ behavioral intention towards heritage protection, OUV attraction had a significant positive impact on tourists’ perceived value (β equal to 0.87) and destination attachment (β equal to 0.85), and played a strong role (route coefficient between 0 to 0.1 is weak, 0.1 to 0.5 is means, and 0.5 to 1 is powerful effect [106]), indicating that the WHS OUV attraction had a significant impact on visitor experience value and emotional engagement, and can transform tourists’ inner states; this further verified the effect of OUV as an important stimulator of tourist behavior [18,20,30,31,33]. Tourists’ heritage education and knowledge had a positive impact on attitudes towards heritage preservation (β equal to 0.33) and intention (β equal to 0.31), signifying the importance of heritage conservation education as a stimulus [37,40,41,42]. At the same time, in the research model, we found that OUV attraction perception and heritage protection education had a significant positive relationship (β equal to 0.61), further demonstrating the intimate association between the two stimuli, and jointly improved tourism value experience, destination attachment, and behavioral intentions to protect heritage. It can be concluded that OUV attraction and heritage protection education had a catalytic impact, further stimulating the inner state and cognition of tourists.

#### 5.1.2. Effect of Tourism Value and Destination Attachment

Tourists’ perceived value had a significant effect on heritage protection attitude (β equal to 0.49) and intention behavior (β equal to 0.20) (medium effect; attitude was more than intentional behavior, indicating that shaping behavior was more difficult than changing attitude). This was consistent with the conclusions that perceived that tourism value had a positive influence on environmentally responsible behavior, tourist satisfaction, and heritage protection behavior [23,82,83,107]. On the other hand, the hypothesis that destination attachment played a significant role in tourist behavioral intention towards heritage protection was not supported in this study. However, most relevant studies showed that tourist place attachment had significant and positive effects on environmental, resource, and heritage protection [23,83]. Some related research report that the effect of destination attachment on visitors’ behavioral intentions is not always positive; it can be positive or negative [108].

The place explained the correlation among the tourism destination and duration of stay, the relationship between the tourist and the travel destination was shallow, and the emotional association was weak, resulting in low attachment to the travel destination [109]. In summary, as the two dimensions of organism in the S-O-R framework, tourism value perception was closely related to cognitive psychological experience, and destination attachment to emotional psychological experience. Moreover, the extent to which destination attachment affected heritage protection attitude and behavior may be related to the length of stay, degree of cognition of site, and preferences of tourists.

#### 5.1.3. Effect of Heritage Conservation Attitude and Intentional Behavior

On the basis of tourists’ OUV attraction perception, heritage protection education, tourism value perception, and destination attachment, it was found that the tourists were stimulated by the outside world, after which they underwent internal state changes, and organically transformed these into a response that valued heritage protection in the S-O-R framework (heritage protection attitude had a significant impact on behavioral intention (β equal to 0.54) [49]). In the TPB, attitudes can influence and even change behavioral intentions [43], and can also be affected by related personal values, beliefs, demographic individualities, and other factors [22,44]. Ramkisson et al. reported that environmental behaviors positively affected place attachment, satisfaction enhanced environmental intentions, which increased place attachment [110], designating that prehistory and aftermath could also experience some extent of conversion and transformation. Tourists’ active practice of heritage conservation behavior may promote their perception of the attractiveness of WHSs’ OUV. To sum up, tourists’ attitude towards heritage conservation is a system of cause and effect under the S-O-R framework, the consequences of which were further promoted by the previous findings to some extent (Figure 3).

### 5.2. Theoretical and Managerial Implications

From a theoretical point of view, the research based on the attitude behavior theory adopted the S-O-R framework, and focused on the changes in tourists’ internal state under the influence of external stimuli and its impact on heritage protection behavioral intention. To some extent, the findings improve the relatively single and decentralized literature on tourists’ heritage protection. We paid more attention to perceived OUV and its impacts on the internal state of tourists, particularly focusing on the dimension of heritage protection education, as it can better stimulate the tourists’ emotional changes. It can verify and even increase the breadth and depth of its theoretical coverage. The role of tourism value perception in heritage protection in the organism dimension was positive, which was coherent with the conclusions of previous studies. However, the influence of destination attachment on protection intention was not significant, indicating that the impact of place attachment theory was not always clear. Place attachment need not necessarily have a positive influence. In the meantime, this study outline has provided new perspectives for further studies on the observation of conservation behavior, which is an extension of the earlier theory of tourist environmental behavior.

From a managerial standpoint, tourists should be paid sufficient attention in their roles as heritage protectors. Tourists’ roles and status in the context of ‘the guest became the host’ should be highlighted, display and publicity of the OUV should be promoted, and channels of heritage protection education should be expanded (such as by effectively interpreting WHS, etc.). In the meantime, to better protect the value of heritage, related progressive technologies such as smart tourism, artificial intelligence, big data, virtual reality, etc., should be introduced and their technological advantages before, during, and after tourist journeys should be furthered [111], thereby enabling tourists to have more channels and choices in heritage site protection, tourism experience, perceived OUV, and destination attachment. In order to better express the degree of OUV feeling and increase tourist satisfaction, we also need to improve tourist ancillary services, such as the accessibility of a scenic resort, the tourist interpretation system, and the service quality of accommodation and food. On the other hand, it is also important to coordinate and balance the benefits of other stakeholders in WHSs; not only should economic interests be considered but the maximum number of tourists should also not be limited. It is necessary to fully consider the ecological, social, and tourist-psychological capacities of heritage sites, taking into account the negative social, economic, and cultural aspects of heritage tourism. We should strive to convince indigenous peoples to protect heritage sites, accommodate and welcome tourists, and support heritage tourism development to directly or indirectly enhance visitor satisfaction and destination attachment.

### 5.3. Limitations and Future Directions

Though the investigation and research have revealed useful findings, there were some shortcomings and scope for further improvement. First, the value of OUV was the essence attractiveness, and an important carrier and stimulus for tourists’ value perception and emotional attachment, which further optimized the OUV for the tourists’ perception measurement scale, considering ancillary attraction and its impact on tourism value perception. Second, the perceived value and destination attachment dimension in organism dimensions can be interpreted from the perspective of tourist satisfaction and environmental behavior in order to improve tourists’ heritage protection behavioral intention. Third, the hypothesis of destination attachment to behavioral intention in this study was not supported; however, it contributed to the literature on place attachment, which can now be expanded to include heritage attachment to study the influence of heritage attachment on heritage protection intention, so as to accurately measure the internal changes and mechanisms involved in tourist emotion. Fourth, based on the S-O-R theory, we studied the behavioral intentions of different stakeholders such as tourists, community residents, tourist practitioners, and heritage managers on heritage protection, identified their common points and differences, revealed their respective influence factors and mechanisms to better formulate the heritage protection policies and programs, and provided a theoretical basis for formulating sustainable development strategies in WHSs. After all, this study was undertaken according to the Chinese cultural context and traditional philosophical think, and therefore had its own characteristics; for example, Taoist thoughts, which have a great influence on Chinese culture, teach that people should respect and adapt to nature (道法自然, follow the law of the development of things themselves.). In addition, the MSNP research region was a Taoist sacred site, which was greatly impacted by Taoist culture; those philosophical thoughts that affected the intention of tourists to preserve heritage were worth examining especially. It was possible to combine the notions of Heaven, the People, and the Land, supporting the concepts of harmony (**天人合一,** unity of Heaven and Human.) and chastity preferences (爱屋及乌, loving someone and caring for someone or something related to that person by association.) from classic Chinese philosophy, to reveal a lot about heritage conservation intention and the environmental behavior of heritage protection intention and environmental behavior, and to compare the relevant study results with those based on the Western cultural context, which may help us to draw more convincing and universal conclusions. In future studies, we should also consider the effects of destination visitor congestion, service quality, visitor satisfaction, and unexpected crisis events (e.g., COVID-19 epidemic) on visitor attitudes and behaviors, which in turn may affect visitors’ willingness and actions for heritage conservation.

## 6. Concluding Remarks

According to the S-O-R outline, this paper studied the relationship and influencing factors of the attitudes and behaviors of tourist heritage conservation in WNHSs, validated the relevant research hypothesis, and discussed the proposed conceptual models. First, OUV attraction of WHSs, heritage education, and tourists’ knowledge were important stimuli for tourists’ heritage protection; OUV attraction perception played a particularly important role in changing tourists’ intrinsic states. Second, the touristic value perceived by tourists had a positive impact on heritage preservation. However, although the assumptions of destination attachment were not supported in this study, the inherent emotional effects of such human–land relationships may be beneficial for heritage protection, indicating the complex transformation scenarios of the two dimensions of organism. Third, the stimulation factors and internal state organism affected the tourist intention of world heritage protection, and heritage protection attitude had a positive effect on behavioral intention, which further proved the explanatory power and appropriateness of the S-O-R framework. Fourth, this research expands the frontiers of heritage conservation study and provides certain theoretical and practical guidelines. It also presents a new viewpoint on theoretical study and development practices for protecting heritage sites and sustainable tourism development, as well as promotes the safeguard and sustainable development of WHSs.

## Figures and Tables

**Figure 1 ijerph-20-01189-f001:**
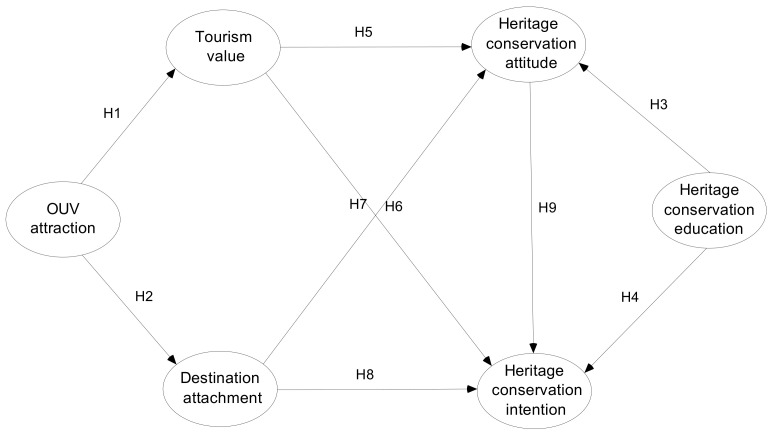
The proposed conceptual model.

**Figure 2 ijerph-20-01189-f002:**
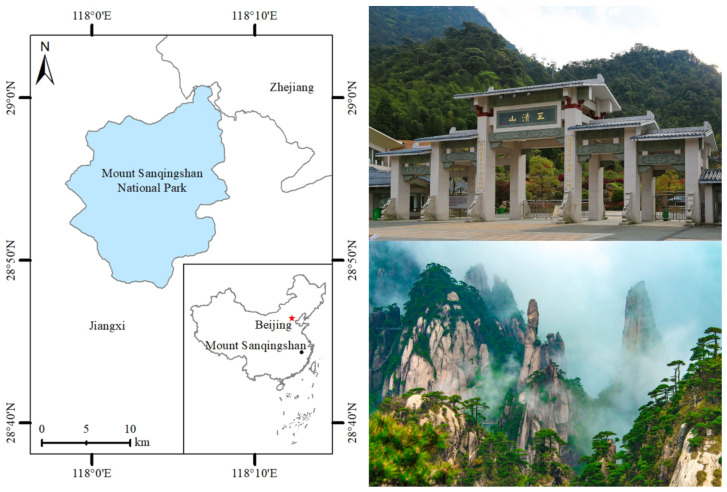
The geographical location and panoramic view of MSNP.

**Figure 3 ijerph-20-01189-f003:**
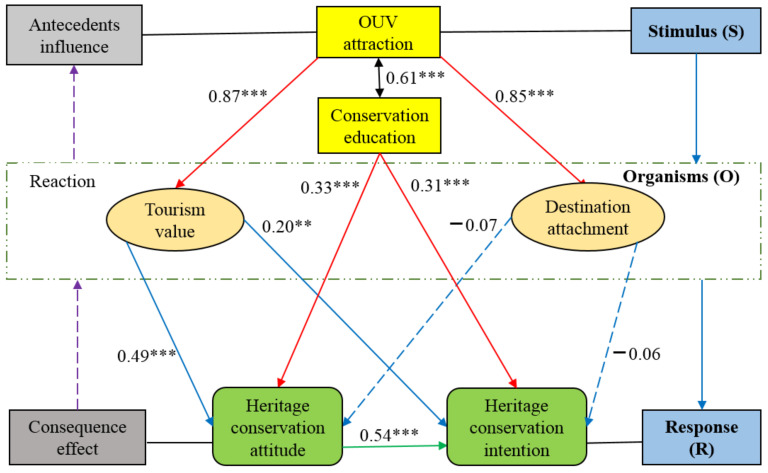
Results of structural model testing and the S-O-R framework. Note: significance, ** *p* < 0.01, *** *p* < 0.001; the solid line is the significant route, but the blue dotted line is not.

**Table 1 ijerph-20-01189-t001:** Measurement elements.

Constructs		Items
**OA**		OUV attraction
	OA1	MSNP’s natural scenery and landscape is mesmerizing.
	OA2	MSNP’s magnificent peaks and rocks are amazing.
	OA3	MSNP’s canyons and clouds are intoxicating.
	OA4	MSNP’s vegetation and ecology are comfortable.
	OA5	MSNP’s high-altitude suspended walking trails are awesome.
CE		Heritage conservation education.
	CE1	The Tourism Interpretation System gives me some knowledge about heritage protection.
	CE2	I have paid attention to related programmers on heritage protection.
	CE3	I am interested in popularization of scientific education for heritage protection.
	CE4	I am very concerned about the knowledge of heritage protection.
TV		Tourism value.
	TV1	This tour has improved my geological and geomorphological knowledge.
	TV2	This tour has achieved my goal of integrating into nature.
	TV3	This tour has satisfied my desire to explore magnificent peaks and rocks.
	TV4	This tour has enhanced my awareness of ecological protection.
	TV5	This tour made me experience the value of WHSs.
DA		Destination attachment.
	DA1	I feel MSNP is more satisfied than other scenic resort.
	DA2	I have a deep place identity to MSNP.
	DA3	I am full of attachment to MSNP.
	DA4	I have an unforgettable trip to MSNP.
CA		Heritage conservation attitude.
	CA1	The tourists’ understanding of the aesthetic value of MSNP should be enhanced.
	CA2	The promotion and education of visitors’ heritage preservation should be strengthened.
	CA3	MSNP’s magnificent natural heritage landscape should be protected.
CI		Heritage conservation intention.
	CI1	I will cherish and appreciate the ecological environment.
	CI2	I will adhere to the multiple management approaches of heritage preservation.
	CI3	I will take legal proceedings to prevent to destruct heritage act.
	CI4	I will prevent the behavior that is harmful to heritage conservation.
	CI5	I will positively contribute in multiple heritage conservation actions.

Note: OA, OUV attraction; CE, conservation education; TV, tourism value; DA, destination attachment; CA, conservation attitude; and CI, conservation intention.

**Table 2 ijerph-20-01189-t002:** Sample characteristics (N = 563).

Items		%			%
Gender	Male	45.2	Income (RMB /month)	≤1500	37.8
	Female	54.8		1501–3500	24.0
Age	≤20	32.4		3501–5000	22.0
	21–30	26.2		5001–8000	10.5
	31–40	25.4		≥8001	5.7
	41–50	11.9	Educational level	Middle school or below	18.9
	≥51	4.1		High school and college graduate	43.4
Travel manner	Tour group	33.7		University graduates and above	37.7
	Family or friends, etc.	41.2	Profession	Enterprise personnel	19.5
	Organizational travel	11.1		Professionals, civil servants	14.6
	Travel alone	4.9		Business services staff	8.5
	Others	9.1		Worker, farmer	5.0
Tourism purposes	Natural sightseeing	72.3		Student	39.8
	Leisure vacation	18.0		Others	12.6
	Others	9.7			

**Table 3 ijerph-20-01189-t003:** Means, reliability, and convergent validity of items.

Variables	Items	Mean	SD	Standardized Loading	T-Value	Cronbach’s Alpha	AVE	CR
OA	OA1	4.39	0.79	0.79	20.94	0.90	0.60	0.88
	OA2	4.42	0.80	0.77	16.97			
	OA3	4.31	0.86	0.75	16.34			
	OA4	4.36	0.81	0.78	16.70			
	OA5	4.47	0.78	0.69	17.22			
CE	CE1	3.94	1.03	0.69	15.52	0.86	0.58	0.85
	CE2	3.94	1.01	0.84	19.33			
	CE3	3.79	1.12	0.78	16.51			
	CE4	3.77	1.10	0.73	21.98			
TV	TV1	4.07	0.96	0.75	18.29	0.89	0.62	0.89
	TV2	4.26	0.87	0.82	19.66			
	TV3	4.13	0.98	0.80	19.10			
	TV4	4.22	0.91	0.79	18.87			
	TV5	4.43	0.81	0.79	17.32			
DA	DA1	3.95	1.0	0.77	21.92	0.89	0.68	0.89
	DA2	4.05	0.97	0.88	20.98			
	DA3	3.83	1.04	0.82	21.11			
	DA4	4.19	0.89	0.81	19.40			
CA	CA1	4.38	0.83	0.86	24.54	0.86	0.68	0.86
	CA2	4.27	0.88	0.76	20.13			
	CA3	4.42	0.84	0.85	24.06			
CI	CI1	4.43	0.81	0.77	21.39	0.86	0.57	0.87
	CI2	4.28	0.85	0.76	20.64			
	CI3	4.18	0.92	0.77	18.87			
	CI4	4.21	0.87	0.75	17.97			
	CI5	3.93	1.06	0.73	17.40			

Note: OA, OUV attraction; CE, conservation education; TV, tourism value; DA, destination attachment; CA, conservation attitude; and CI, conservation intention.

**Table 4 ijerph-20-01189-t004:** Discriminant validity.

Items	OA	CE	TV	DA	CA	CI
OA	0.775					
CE	0.611	0.762				
TV	0.754	0.529	0.787			
DA	0.651	0.520	0.737	0.825		
CA	0.563	0.549	0.611	0.460	0.825	
CI	0.616	0.683	0.650	0.498	0.703	0.755

Note: OA, OUV attraction; CE, conservation education; TV, tourism value; DA, destination attachment; CA, conservation attitude; and CI, conservation intentions.

**Table 5 ijerph-20-01189-t005:** Goodness-of-fit directories.

Model-Fit Directory	Absolute Index			Comparative Index			Parsimony Index		
	CMIN/DF	GFI	RMSEA	IFI	TLI	CFI	PGFI	PNFI	PCFI
Threshold values	2–5	>0.90	<0.08	>0.90	>0.90	>0.90	>0.50	>0.50	>0.50
Theoretical model	4.4	0.845	0.078	0.899	0.887	0.899	0.698	0.780	0.802
Revised model	3.2	0.879	0.063	0.937	0.927	0.937	0.704	0.788	0.810

## Data Availability

No data available to share at this time.

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
