# Peer review of "Stimulus-Organism-Response Framework: Is the Perceived Outstanding Universal Value Attractiveness of Tourists Beneficial to World Heritage Site Conservation?"

_ijerph, 2023, doi:10.3390/ijerph20021189_

Round 1

Reviewer 1 Report

Thanks for giving the opportunity to review the paper titled as Stimulus-organism-response framework: Is the perceived outstanding universal value attractiveness of tourists conductive to world heritage site conservation?. This paper is timely, well developed, conducted and well written. It addresses a significant topic likely to be of interest to world heritage destination sustainable development and conservation. Thus, it addresses a significant topic likely to be of interest to International Journal of Environmental Research and Public Health. Despite all of this, there are several possible revisions as the following:

First, The data were collected in August 2015. However, the COVID-19 pandemic as typical destination crisis event busted, and has greatly influence on tourist crisis perception, specially has negative impacts on tourist’s emotion, attitudes and behavior in turn influence sustainable tourism development (Kimbu, Adam, Dayour, & Jong, 2023; Su, Jia, & Huang, 2022; Su, Pan, & Huang, 2023; Yang, Zhang, & Chen, 2020). Thus, please author(s) refer(s) these references, and explain what they do think of this impact on the data. At least, author(s) need mention them, and list these as a limitation.

“How do destination negative events trigger tourists’ perceived betrayal and boycott? The moderating role of relationship quality”. Tourism Management, 92, 104536.

“How does destination crisis event type impact tourist emotion and forgiveness? The moderating role of destination crisis history”. Tourism Management, 94, 104636. 

“COVID-19-Induced redundancy and socio-Psychological well-being of tourism employees: Implications for organizational recovery in a resource-scarce context”. Journal of Travel Research, 62(1), 55-74.

“Coronavirus pandemic and tourism: Dynamic stochastic general equilibrium modeling of infectious disease outbreak”. Annals of Tourism Research, 83, 102913.

Second, it needs point out the research gaps more clearly in the introduction section.

Third, the theoretical model of SEM needs more clearly. Specially, what’s the role of destination attachment, heritage conservation attitude, heritage conservation education, and the relationships among them.

Fourth, overall, the contribution of the paper and the discussion of the contribution should be made more clearly.

Fifth, it need point out the limitations and future research directions more specifically in the end of the paper.

Though the above-mentioned possible shortages, this paper is a high-quality original article. Thus, I recommend author(s) revise(s) the paper according to reviewer’s comments/suggestions.

Reviewer 2 Report

The title and the aim of the paper are informative and relevant. However, without being an expert in English I rather consider that word “conducive” can be replaced from the title. The abstract is descriptive and informative about the aim of the paper and results are presented in extenso. I suggest authors to include some words about the methodology of the research in the abstract.

The Conceptual Framework and Hypotheses are presented very clear and detailed. The research questions are clearly outlined. The only observation for this part is related with the fact that I searched for the travel resort charm” (lines 178-180) definition in the reference 19 article and I did not find any mention.

Methodology part of the study is very detailed and the study area is described. As the authors gave some data about the number of visitors and revenues from 2002 (prior to WNHS title award in 2008) and 2019 (after WNHS title award), I suggest a short comment and some relevant references about the possible influence of WNHS award, AAAAA tourist area national top-level tourist destination classification in 2011, or World Geopark status given in 2012.

The table 1 is comprehensive and measurement instruments were created based on extensive literature. From this perspective, the authors provided details for future researchers interested in replication of the study.

The results are presented in an appropriate way. Conclusions are related with aims of the study and presented clearly. The limitations of the study are stated and the future directions invite future researchers from different cultural and geographical contexts.
